# Effect of Acupuncture in Pain Management of Head and Neck Cancer Radiotherapy: Prospective Randomized Unicentric Study

**DOI:** 10.3390/jcm10051111

**Published:** 2021-03-07

**Authors:** Radana Dymackova, Iveta Selingerova, Tomas Kazda, Marek Slavik, Jana Halamkova, Michaela Svajdova, Pavel Slampa, Ondrej Slama

**Affiliations:** 1Department of Radiation Oncology, Faculty of Medicine, Masaryk University, 625 00 Brno, Czech Republic; radana.dymackova@mou.cz (R.D.); slavik@mou.cz (M.S.); svajdovam@uvn.sk (M.S.); 2Department of Radiation Oncology, Masaryk Memorial Cancer Institute, 656 53 Brno, Czech Republic; tomas.kazda@mou.cz (T.K.); slampa@mou.cz (P.S.); 3Research Center for Applied Molecular Oncology (RECAMO), Masaryk Memorial Cancer Institute, 656 53 Brno, Czech Republic; 4Department of Comprehensive Cancer Care, Faculty of Medicine, Masaryk University, 625 00 Brno, Czech Republic; jana.halamkova@mou.cz; 5Department of Comprehensive Cancer Care, Masaryk Memorial Cancer Institute, 656 53 Brno, Czech Republic; oslama@mou.cz; 6Clinic of Radiation and Clinical Oncology, Central Military Hospital—Teaching Hospital, 034 01 Ruzomberok, Slovakia

**Keywords:** acupuncture, head and neck cancer, radiotherapy, toxicity, pain

## Abstract

This prospective randomized open-label trial aimed to evaluate the role of acupuncture in the treatment of pain related to curative and adjuvant (chemo)radiotherapy of head and neck cancer. Patients in two arms (30 patients in each arm) underwent standard oncology therapy and standard supportive care with or without acupuncture. The stratification factors were the type of treatment and chemotherapy indication. The toxicity assessed was represented by pain rated on a 10-point pain scale and analgesic use. Average pain (AP) and the worst pain during the day (WP) were significantly lower in the acupuncture arm during radiotherapy (AP median 0.16 vs. 1.36, *p* < 0.001; WP median 0.90 vs. 1.96, *p* < 0.001) and three months after radiotherapy (AP median 0.07 vs. 0.50, *p* < 0.001; WP median 0.30 vs. 0.83, *p* = 0.002). The analgesic consumption between arms was statistically significantly different. A median of the proportion of days when the patients used analgesics was 8% and 32.5% during radiotherapy (*p* = 0.047) and 0% and 20.8% during three months after radiotherapy (*p* = 0.006) for the acupuncture and control arm, respectively. Results point out lower analgesic consumption and milder pain in acupuncture arm. Acupuncture consequently offers another alternative to standard treatment leading to a reduction in the toxicity of oncological treatment.

## 1. Introduction

The treatment of head and neck cancer is very challenging. The main methods used in curative therapy of head and neck cancer include surgery and radiotherapy or chemoradiotherapy [1,2,3]. Radiotherapy (RT) is offered as a curative method for inoperable diseases or if surgery is not indicated as well as an adjuvant method after surgery in patients with high-risk factors of local recurrence [4]. Oncologic therapies of these patients are often accompanied by both acute and chronic adverse effects. The main acute adverse effects include radiodermatitis and radiostomatitis, xerostomia, loss of taste, and also pain and nausea. Severe acute toxicity may alter adherence to complete curative treatment as prescribed and, thus, impact overall survival. The adverse effects may become chronic, particularly the loss of saliva and taste, as well as cutaneous and mucosal toxicity and pain. Consequently, in addition to treating the disease itself, it is also essential to treat the adverse effects of oncological treatment. Nowadays, there is no possibility to cure the adverse effects of RT effectively, and management of these symptoms is purely symptomatic. Acupuncture is one of the methods which may be very helpful in this clinical situation [5,6].

The use of acupuncture in the treatment of pain in cancer patients has proven useful in several studies [7,8]. The positive results of these studies are also reflected in the NCCN (National Comprehensive Cancer Network) recommendations that acupuncture is a more promising approach supported by scientific plausibility, although the technique lacks standardization. It is noninvasive and relatively inexpensive, and it may be considered as an adjunct option in treating patients with medication-resistant chemotherapy-induced peripheral neuropathy [9].

We conducted a single institutional prospective open-label randomized study (NCT03751566) focused on the use of acupuncture as a treatment modality for adverse effects of radiotherapy or chemoradiotherapy in patients with head and neck cancer [10]. The study was approved by Ethical Board of Masaryk Memorial Cancer Institute.

This article presents the results of the assessment of pain grade during oncologic treatment and the usage of analgesics. The study also evaluated other acute side effects (skin and mucous toxicity, xerostomia and perception of taste, and nausea), the consumption of antiemetics and chronic toxicity. The results of these evaluations will be part of other separated reports.

## 2. Materials and Methods

### 2.1. Study Design, Population, and Randomization

This prospective single-institution study was conducted in the Masaryk Memorial Cancer Institute in Brno, Czech Republic. The study duration was two years; patients with head and neck cancer indicated to curative or adjuvant radiotherapy were enrolled from March 2016 to June 2018. A total of 439 patients was evaluated for eligibility. Inclusion criteria were age > 18 years, Karnofsky performance status over 60%, physical and intellectual ability (as assessed subjectively by the investigator) to manage the therapy diary for toxicity records. Exclusion criteria were enrollment in another clinical trial and palliative intent of the treatment with a lower prescribed dose of RT. Sixty-two eligible patients agreed with enrollment. The patients were randomized to control or intervention arm by permuted stratified block randomization (block size 4). The stratification factors included the type of RT (curative/adjuvant) and the application of concurrent chemotherapy (yes/no). Two patients randomized into the control arm changed opinion with participation in the study immediately after randomization, were excluded and replaced by others to achieve a total number of 60 participants. A CONSORT diagram detailing patient flow is presented in Figure 1. Informed consent was obtained from each participant.

All patients finished the treatment as planned according to the protocol. No deaths or serious adverse events were reported during the study. Two patients (one in each randomized arm) were lost from follow-up and did not have any control examination after the termination of treatment.

The primary endpoint was reducing acute toxicity in acupuncture arm assessed from the perspective of pain with the pain level on the Likert scale (see below) as the endpoint. Reduction of usage of analgesics and antiemetics, acute toxicity of nausea, skin and mucous membrane, and chronic toxicity were secondary outcomes.

Trial Registration: Clinicaltrials.gov accessed on 6 March 2021—NCT03751566.

### 2.2. Standard Treatment

The patients in both arms underwent standard treatment. It included curative or adjuvant radiotherapy with concurrent chemotherapy in some patients. According to the nature of treatment and the risk factors involved, radiotherapy of up to 60–70 Gy was applied. A single dose was either 2 Gy or 1.6 to 2.12 Gy when a simultaneous integrated boost technique was applied. The whole treatment usually lasted 6–7 weeks. Radiotherapy was administered via a linear accelerator using the RapidArc dynamic delivery technique.

Chemotherapy was administered to selected patients (10 in the acupuncture arm and 8 in the control arm) upon the assessment of risk factors. The most common concomitant chemotherapy regimen was cisplatin administrated weekly (30–40 mg/m^2^) or on days 1, 22, and 43 (100 mg/m^2^). Two patients in whom cisplatin was contraindicated had a regimen with weekly-administered carboplatin (AUC 1.5). Two patients received radiotherapy with concomitant targeted therapy cetuximab.

Standard supportive therapy included local treatment of skin and mucosal reactions [10]. Analgesic and antiemetic therapies were given as indicated. Analgesic therapy was administered according to the degree of pain. Thus, the therapy ranged from non-opioids (metamizole and paracetamol) through weak opioids (tramadol or codeine) to strong opioids (fentanyl, buprenorphine, or morphine).

### 2.3. Acupuncture

Acupuncture was carried out by a certified physician (R.D.—two-year course, basics of traditional Chinese medicine (TCM)—diagnostician TCM + one-year course, The science of acupuncture points and pathways at the Academy of TCM in Brno; one-year course, basic course of acupuncture in Institute for Postgraduate Medical Education in Prague; finished with an exam and received a certificate). As per the study protocol, it was initiated upon the first symptoms of toxicity of the oncologic treatment. During radiotherapy, acupuncture was applied 1–3 times per week, usually twice per week for 10–20 min. After the end of the radiotherapy period, acupuncture was applied usually once in 14 days over the period of 1–2 months, in some patients only once a month. After the end of radiotherapy, the frequency of acupuncture application depended on the stage of development of acute toxicity and the patient’s ability to attend the application. The application was individualized; at the beginning of each acupuncture session, the evaluation of radial pulses and the tongue was done to identify appropriate acupoints; points LI4, LU7, ST36, KI7 were mostly included. A whole-body acupuncture approach was applied, except for the irradiated skin, including ear acupuncture.

### 2.4. Pain Assessment

Two pain parameters were assessed: average pain during the day and the worst pain during the day. The patients rated their pain and recorded it in their toxicity diaries using a 10-point pain scale (general Likert scale with graphic and verbal explanations clarifying the meaning of the individual points), where 0 was no pain, and 10 was the worst pain the patient could imagine. Patients filled in the toxicity diary daily or when the condition changed.

### 2.5. Statistical Methods

Patient and treatment characteristics were described using standard summary statistics, i.e., median and interquartile range (IQR) for continuous variables and frequencies and proportions for categorical variables and differences between arm were compared using Fisher exact test or Mann–Whitney test as appropriate. Due to the comparability of the toxicity progression between individual patients, particularly considering the different lengths of RT, time was rescaled to a uniform scale. The timeline is expressed in percentages during RT period or days during three months following the termination of RT. The area under the curve (AUC) of the time course of toxicity was calculated for each patient and standardized on a unit scale. The AUC value shows the average toxicity for each patient per unit of time, i.e., one percent of the RT period or one day after the end of RT. The toxicity parameters were evaluated separately in some critical moments of the treatment and post-treatment period—the start, the mid-point, the three-quarters and the end of RT, 14 and 30 days after the end of RT. The analgesics’ usage was evaluated using the proportion of patients receiving analgesics at a considered moment. The statistical comparisons between control and acupuncture arms were accomplished by the Mann–Whitney test and displayed using box plots. All statistical analyses were performed employing R version 4.0.2 [11], and a significance level of 0.05.

## 3. Results

### 3.1. Patient Characteristics

A total of 60 patients was enrolled in this single-institutional prospective study (30 in each arm). Due to stratification, the proportion of patients undergoing curative and adjuvant therapy was similar in both arms (35% and 65%). Likewise, patients who received concomitant chemotherapy, were equally represented in both arms (approximately 70%). The study included mostly men (77%). The patients’ median age was 59 years. In the acupuncture arm, patients were statistically significantly younger (*p* = 0.026). In the control arm, the majority of patients were current or former smokers (83%). In the acupuncture arm, patients with advanced cancer stage slightly predominated (*p* = 0.088). The occurrence of different severity types of comorbidities was similar in both treatment arms. Demographic and treatment characteristics of study patients are summarized in Table 1.

### 3.2. Pain Assessment

The patient’s pain level expressing the average pain and the worst pain during the day increased during RT. The increase was more pronounced in the control arm. The patients with acupuncture applied during oncologic therapy had a milder progression of pain. After completion of RT, the pain level returned to pre-radiotherapy levels within approximately two months. The time courses of the summary statistics (maximum, mean, and median) for both pain parameters are shown in Figure 2A,B. The difference between arms appears mainly in medians and means. The maximums are similar in both groups. In other words, the acupuncture group also contained patients who assessed the pain they perceived during RT as severe—up to the level of 8 on the 10-point scale. The average pain level per time unit was statistically significantly lower in the acupuncture arm for the average (median 0.16, IQR 0.00–1.09 vs. median 1.36, IQR 0.74–2.03, *p* < 0.001) and the worst (median 0.90, IQR 0.46–1.33 vs. median 1.96, IQR 1.58–2.62, *p* < 0.001) pain during RT, and for the average (median 0.07, IQR 0.00–0.32 vs. median 0.50, IQR 0.24–1.00, *p* < 0.001) and the worst (median 0.30, IQR 0.09–0.53 vs. median 0.83, IQR 0.40–1.16, *p* = 0.002) pain during three months following the termination of RT (Figure 2C).

A separate evaluation at critical moments of the treatment was performed to demonstrate the influence of acupuncture. Specifically, at the start, in the middle, in the three-quarters and at the end of RT, then 14 days after and one month after the end of RT. The patients in the acupuncture arm reported a statistically significant lower pain level in all monitored moments except the start of RT (Figure 3A). The differences in most of the studied moments amount to 1–2 grades in favor of the acupuncture arm, both concerning the average pain and the worst pain during the time unit.

A proportion of days with pain level above three on a 10-grade scale was analyzed for each patient (Figure 3B). Except for a few patients, the number of days with pain level above three was 0 in the acupuncture arm. On the other side, in the control arm, a median of 1% (IQR 0–20%) of the average pain days and 21% (IQR 12–38%) of the worst pain days was observed during the course of RT, respectively. The difference between acupuncture and control arms in the proportion of days was statistically significant for both average and the worst pain during RT (*p* = 0.001 and *p* < 0.001) and three months after the end of RT (*p* = 0.004 and *p* = 0.003).

### 3.3. Evaluation of Analgesic Therapy

Another objective of this study was to evaluate the need for analgesics during RT and whether analgesics’ usage was lower in the acupuncture arm patients. In total, analgesics’ usage in the acupuncture arm was lower both during RT and during three months after the end of RT (Figure 4A). The proportion of patients requiring analgesic therapy at the key moments during the treatment are summarized in the table in Figure 4B.

A median of the proportion of days when the patients used analgesics was 32.5% (IQR 21.0–68.8%) and 8.0% (IQR 0.0–51.5%) during RT and 20.8% (IQR 0.0–28.7%) and 0% (IQR 0.0–13.9%) during three months after the termination of RT in the control and acupuncture arm, respectively. These differences between arms reached statistical significance *p* = 0.047 during RT and *p* = 0.006 after the end of RT (Figure 4C).

The evaluation also covered the individual groups of analgesics—usage of medicinal products within the individual groups was also lower in the acupuncture arm than in the control arm (Figure 5A). Although the proportion of days when the patient used non-opioids, i.e., of the weak analgesics group, was lower in the acupuncture arm (*p* = 0.161 during RT, *p* = 0.322 during the first three months after RT) than in the control group, the difference observed was not statistically significant. The usage of weak opioids was statistically significantly lower in the acupuncture arm (*p* = 0.022 during RT, *p* = 0.038 during the first three months after RT) than in the control arm (Figure 5B). For strong opioids, the median of the proportion of days of use was 0% for both acupuncture and control arms.

## 4. Discussion

Using acupuncture in the therapy of side effects in oncologic treatment is currently not a standard in the Czech Republic. However, foreign studies report that patients can benefit from acupuncture applied during oncologic therapy. In a recent meta-analysis, a positive effect of acupuncture on cancer-related pain has been proven [12]. Acupuncture can improve patients’ tolerance of therapy as well as shorten the recovery period. Previous clinical studies showed the effect of acupuncture in other radiotherapy-related adverse effects such as xerostomia [13] or dysphagia [14]. This study was driven by the motivation to verify that assumption on a cohort of our own patients. The study evaluated several parameters.

One of the main symptoms against which acupuncture is used and the most well-known indication of acupuncture is pain. This study proved a statistically significant manifestation of the difference in pain assessment, both average pain (*p* < 0.001 during RT as well as three months after the end of RT) and the worst pain during the day (*p* < 0.001 during RT, *p* = 0.002 three months after the end of RT) in favor of the patients enrolled in the acupuncture arm. A United States study by Miller et al. [15] published a similar statistically significant decrease in tumor-related pain in patients receiving palliative care as a result of acupuncture. Kasymjanova et al. [16] present another study focusing on tumor-related pain, which recorded a decrease in pain in 61% of patients when acupuncture was used. A review published by Lian et al. [17] in 2014 discussed six studies dealing with pain therapy used against adverse effects of oncologic therapies. All six studies witnessed a statistically significant difference in pain control.

A randomized controlled trial from 2010 [18] looked at the use of acupuncture for pain and dysphagia in patients after neck dissection. The study included 58 patients divided into two groups. Patients in acupuncture group were treated with acupuncture. Patients in the control group were treated with standard therapy (analgesics, physiotherapy, antirheumatics). The results showed that pain was improved in patients in the acupuncture group according to the Constant Murle score by 14%, in patients in the control group by only 1.5%. Even in the evaluation of xerostomia according to Xerostomia Inventory, a decrease in problems was recorded in patients—in the acupuncture group by 7%, in the control group only by 1.5%. There was also a significant difference in the evaluation of pain on the pain scale—patients in acupuncture group had a decrease in pain of 20%, patients of the control group only 1%.

In another study by Chen et al. [19], patients with advanced tumors suffering from pain were divided into three groups according to pain intensity. Each group was further divided into a subgroup treated with acupuncture (3–5 acupuncture points were used) and a subgroup treated with conventional WHO (World health organization) analgesics (mild pain—aspirin, moderate pain—codeine, severe pain—morphine). The group of patients treated with conventional analgesics had pain satisfactorily controlled in 87.5%. Patients treated with acupuncture alone without analgesic drug treatment had pain satisfactorily controlled in 94.1%. In addition, patients in this group did not experience any side effects associated with conventional analgesic therapy.

Therefore, acupuncture seems to be beneficial in pain therapy, according to our results and foreign studies as well. Our study, as well as studies found in the previous literature, describe the benefit of acupuncture in pain therapy as statistically significant. We confirm the effectiveness of acupuncture on pain as described in current NCCN guidelines [1].

The other objective of this study was lowering the usage of analgesics. Usage of analgesics was statistically significantly lower both during RT (*p* = 0.047) and during three months after the end of RT (*p* = 0.006). The largest difference in the amount used analgesics was observed for the weak opioids which patients use only in more intense pain. This pain was observed mainly in the control group, therefore, consumption of weak opioids was higher. On the other side, non-opioids are commonly used as first-line analgesics even with mild pain. The use of strong opioids was not common and involved very high pains. Concerning the pain maximum was similar in both arms, the consumption of analgesics did not differ either.

A randomized sham-controlled trial [20] from 2020 evaluates acupuncture for symptom management and reduction of pain medication use during cancer treatment in patients with multiple myeloma. Patients were randomized to receive either true or sham acupuncture. Patients who received sham acupuncture were five times more likely to increase pain medications from baseline when compared with patients who received true acupuncture.

Both the intensity of pain during the treatment and analgesics usage were lower in the acupuncture arm during this study. Since pain has a major impact on the quality of patients’ life and their tolerance of the treatment, the results presented here indicate that acupuncture during RT applied to the head and neck area is clinically significant.

We are aware of some limitations of the study. Firstly, patients were stratified into arms according to treatment characteristics, however, other patients’ characteristics are different between arms. Mainly, patients in the control arm are older and more often smokers. On the other side, there are patients with more advanced diseases in the acupuncture arm. All these facts may bias the results. The subjectivity of pain assessment and the impossibility of blinding used intervention are other disadvantages that may be considered when planning further studies.

In conclusion, acupuncture is a well-tolerated, feasible and effective treatment of radiotherapy-related pain in head neck cancer patients and leads to lower usage of mild opioids. The use of acupuncture can have an impact not only on improving the quality of life but also on reducing the side effects and costs of analgesic therapy. Further research is needed for the inclusion of acupuncture in the standard-of-care treatment process in this group of patients.

## Figures and Tables

**Figure 1 jcm-10-01111-f001:**
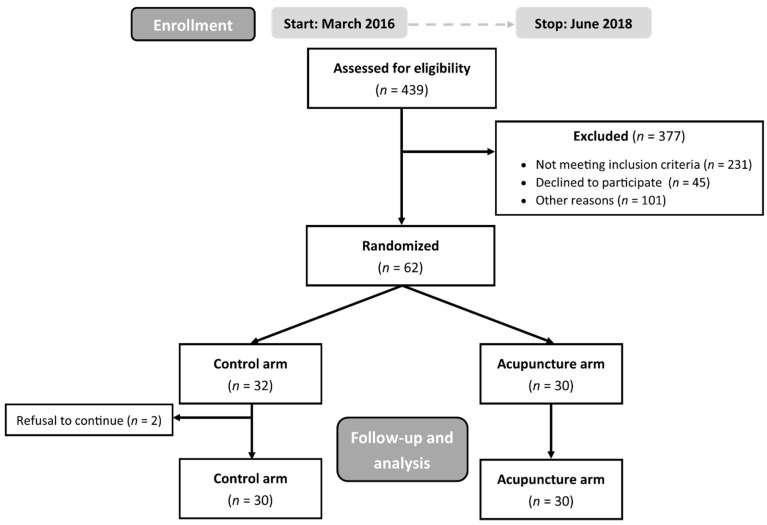
CONSORT flow diagram of the study participants.

**Figure 2 jcm-10-01111-f002:**
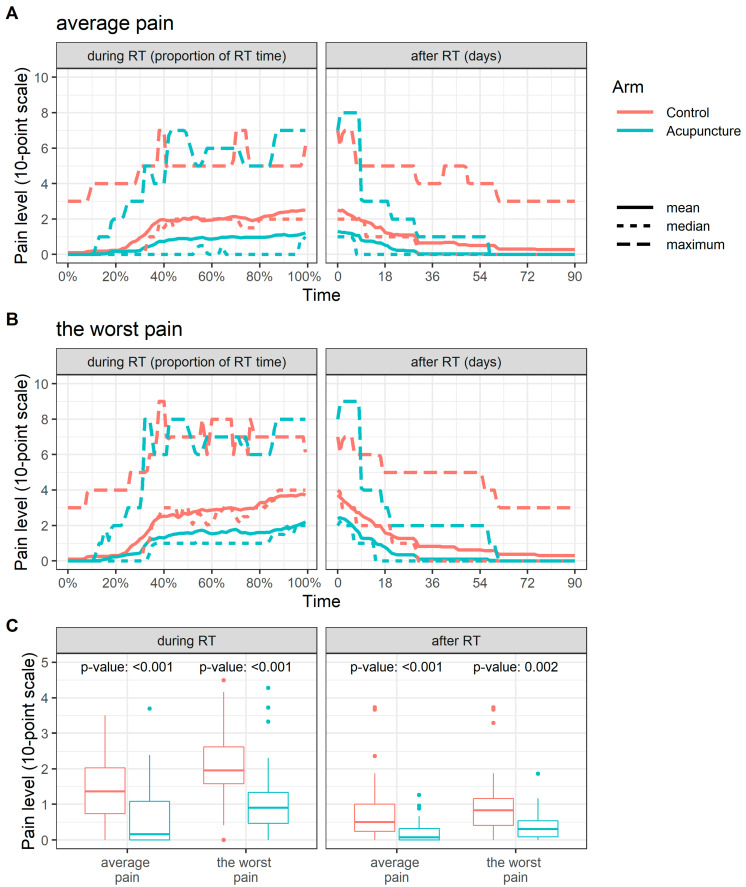
Time course of summary statistics for (**A**) average and (**B**) the worst pain for control (red) and acupuncture (cyan) arm. (**C**) Average pain level per time unit for control (red) and acupuncture (cyan) arm.

**Figure 3 jcm-10-01111-f003:**
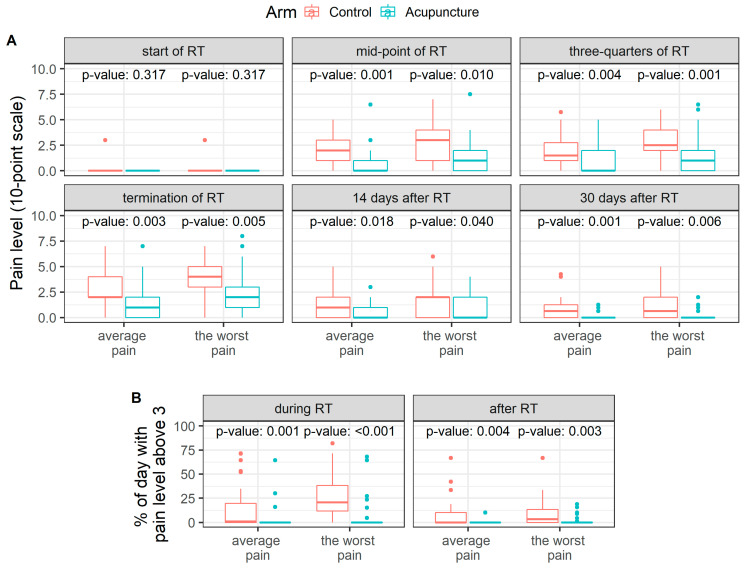
(**A**) Pain level in critical moments of treatment. (**B**) Percentage of days during radiotherapy and three months after radiotherapy when patients had pain level above three on a 10-grade scale.

**Figure 4 jcm-10-01111-f004:**
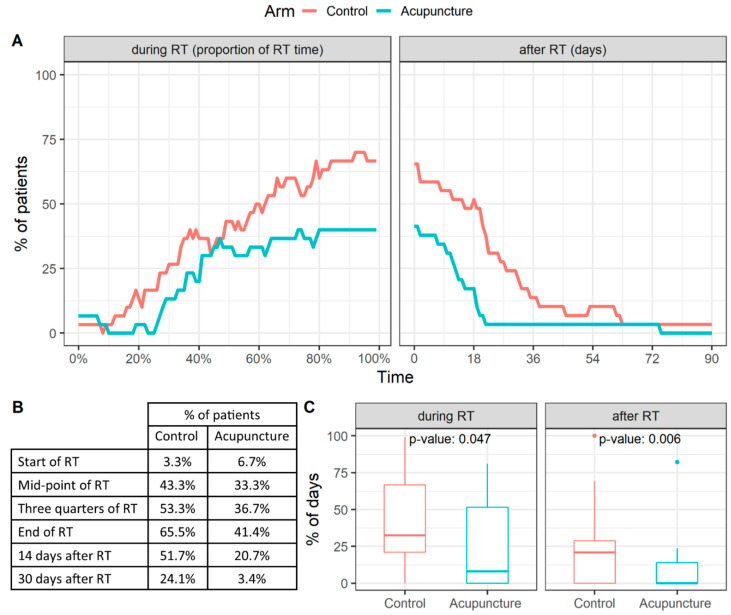
(**A**) Time course and (**B**) the key moments during the treatment of the proportion of patients requiring analgesic therapy. (**C**) The proportion of days when the patients used analgesics.

**Figure 5 jcm-10-01111-f005:**
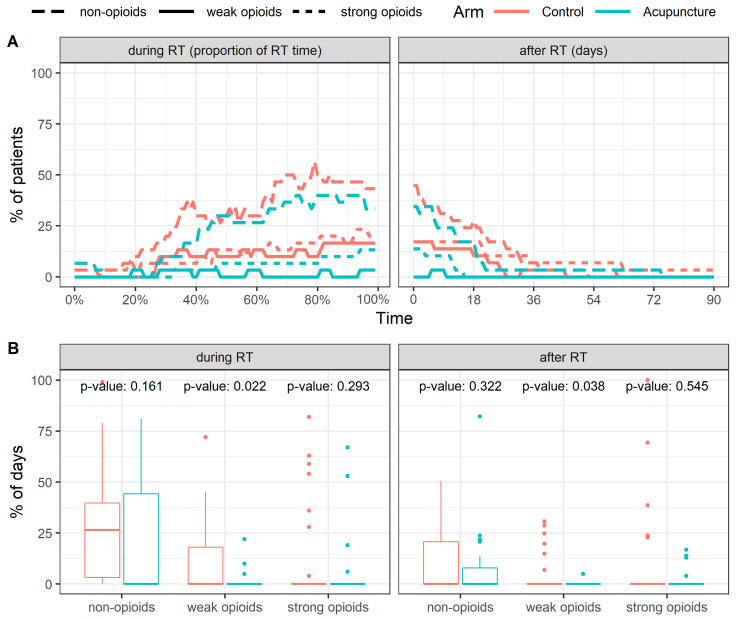
(**A**) Time course and (**B**) the key moments during the treatment of the proportion of patients requiring analgesic therapy divided according to the individual groups of analgesics.

**Table 1 jcm-10-01111-t001:** Demographic and treatment characteristics.

	Total (*N* = 60)	Acupuncture Arm (*N* = 30)	Control Arm (*N* = 30)	*p*-Value
**Sex**				
Male	46 (77%)	22 (73%)	24 (80%)	0.761
Female	14 (23%)	8 (27%)	6 (20%)	
**Age**				
Median (IQR)	59 (49; 67)	51 (47; 64)	61 (55; 68)	**0.026**
**Smoker/Non-smoker**				
Smoker	24 (41%)	8 (28%)	16 (53%)	**0.028**
Former smoker	16 (27%)	7 (24%)	9 (30%)	
Non-smoker	19 (32%)	14 (48%)	5 (17%)	
Unknown	1/60 (2%)	1/30 (3%)	0/30 (0%)	
**Site of tumor**				
Oropharynx	22 (37%)	13 (43%)	9 (30%)	0.370
Larynx	17 (28%)	6 (20%)	11 (37%)	
Oral cavity	8 (13%)	3 (10%)	5 (17%)	
Nasopharynx	2 (3%)	2 (7%)	0 (0%)	
Primum unknown	7 (12%)	3 (10%)	4 (13%)	
Other	4 (7%)	3 (10%)	1 (3%)	
**Stage**				
I	6 (10%)	3 (10%)	3 (10%)	0.088
II	9 (15%)	1 (3%)	8 (27%)	
III	16 (27%)	9 (30%)	7 (23%)	
IV	29 (48%)	17 (57%)	12 (40%)	
**Type of tumor**				
Spinocellular ca	53 (88%)	24 (80%)	29 (97%)	
Other	7 (12%)	6 (20%)	1 (3%)	
**Grading**				
G1	5 (8%)	2 (7)	3 (10%)	0.795
G2, G1–2	26 (43%)	12 (40%)	14 (47%)	
G3, G2–3	18 (30%)	9 (30%)	9 (30%)	
G4	2 (3%)	2 (7%)	0 (0%)	
Not determined	9 (15%)	5 (17%)	4 (13%)	
**HPV status**				
Negative	13 (22%)	7 (23%)	6 (20%)	>0.999
Positive	25 (42%)	12 (40%)	13 (43%)	
Not determined	22 (36%)	11 (37%)	11 (37%)	
**Curative/adjuvant**				>0.999
Adjuvant	39 (65%)	19 (63%)	20 (67%)	
Curative	21 (35%)	11 (37%)	10 (33%)	
**Chemotherapy**				0.779
Yes	18 (30%)	10 (33%)	8 (27%)	
No	42 (70%)	20 (67%)	22 (73%)	
**Volume irradiated with the lowest dose**				
Median (IQR)	598 (330; 751)	677 (344; 756)	463 (294; 719)	0.398
**Volume irradiated with the highest dose**				
Median (IQR)	153 (86; 293)	132 (63; 270)	182 (103; 352)	0.123
**Comorbidities—severity**				
Mild *	25 (42%)	13 (43%)	12 (40%)	0.861
Severe **	22 (37%)	10 (33%)	12 (40%)	
None	13 (22%)	7 (23%)	6 (20%)	

Legend: N—number; IQR—interquartile range; ca—carcinoma; HPV—human papillomavirus; * hypertension, diabetes mellitus without serious complications, hypothyroidism, hyperlipidemia, vertebrogenic algic syndrome; ** ischemic heart disease, myocardial infarction in the anamnesis, brain stroke in the anamnesis, epilepsy, pulmonary embolism in the anamnesis, multiple oncological diagnoses, rheumatic diseases, chronic obstructive pulmonary disease.

## Data Availability

The data presented in this study are available on request from the corresponding author.

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
