# Peer review of "Effect of Acupuncture in Pain Management of Head and Neck Cancer Radiotherapy: Prospective Randomized Unicentric Study"

_jcm, 2021, doi:10.3390/jcm10051111_

Round 1
Reviewer 1 Report
This is an interesting study about the effect of acupuncture in pain management of head and neck cancer radiotherapy. The authors evaluated 60 patients (30 patients in each arm) who underwent standard oncology therapy and standard supportive care with or without acupuncture. Average pain and the worst pain during the day were significantly lower in the acupuncture arm during radiotherapy and three months after radiotherapy. The analgesic consumption between arms was statistically significantly different, being lower in acupuncture arm.
The paper is well written. However, some issues remain.
In the Introduction section, the authors stated that curative radiotherapy is used only for inoperable disease. This is not true, since some tumors, such as T1a cancer of the vocal fold, may undergo surgery or curative radiotherapy. Please correct the sentence.
The Introduction section may be implemented with some data about pain in head and neck radiotherapy and about the role of acupuncture in the treatment of different causes of pain.
In Table 1, the authors must report p values for all the parameters. Moreover, they should report the site of tumors (e.g. oral cavity, oropharynx, larynx, hypopharynx, nasopharynx, paranasal sinuses…).
I think that maximum pain in Figure 1A and 1B is confounding and should be removed.
Reporting the site of pain (skin, throat, other?) may be useful to better understand patients’ problems and the effect of acupuncture.
Was pain higher in patients who underwent higher radiation doses or concurrent chemotherapy?
The Discussion section is too short and should be implemented with more data about acupuncture and about drug use for pain in head and neck radiation.
Reviewer 2 Report
It is meaningful in that it is a clinical study that has seen the effect of acupuncture on pain in radiotherapy in head and neck cancer. However, there are questions about some of the following.The authors noted that the detailed design was published in the 'pilot data report'. However, it is not a pilot data report. Reference paper (7) show that the interim analysis of this study. Reference (7) was reported when the recruitment of 30 people was completed in the 60 recruitment study. The authors wrote "pilot data report", but they need to change the word "interim analysis report" exactly. Was this interim analysis originally planned? If the interim analysis that was not planned is reported, then the novelty of this paper may be judged to be low.
The authors need to describe the research method in more detail. The authors noted that there is a detailed protocol in reference (7), but only the randomization method is described in detail, and neither in the Clinical Trial gov registration nor in previous papers can be detailed about the protocol.
The allocation concealment needs to be described how it is performed, and the randomization method needs to be summarized in this paper. It also requires the details in what primary outcome is and what primary endpoint was. In the registration information for NCT03751566, the time frame for the primary outcome is six months after radiotherapy. However, the authors in this manuscript deals with radiotherapy information after three months and does not mention six months. It is necessary to reveal what the original primary endpoint was.
It is necessary to present the flowchart according to the CONSORT guidelines. There is no mention of drop out or withdrawl.
Did all patients complete the treatment specified in protocol?
Any patients who have died or experienced SAE during the study period should be reported in flow chart.
According to that interim analysis, not only pain toxicity, but also skin toxicity and mucosal toxicity was presented. It is necessary to confirm whether the authors will report on this result in other studies.
Based on the mean value of the average pain (it is also low in control group), some question is whether the pain in the RT of these patients is clinically significant. It is necessary to mention limitations on this.
Author Response
Response to Reviewer 2 Comments
Point 1: It is meaningful in that it is a clinical study that has seen the effect of acupuncture on pain in radiotherapy in head and neck cancer. However, there are questions about some of the following.The authors noted that the detailed design was published in the 'pilot data report'. However, it is not a pilot data report. Reference paper (7) show that the interim analysis of this study. Reference (7) was reported when the recruitment of 30 people was completed in the 60 recruitment study. The authors wrote "pilot data report", but they need to change the word "interim analysis report" exactly. Was this interim analysis originally planned? If the interim analysis that was not planned is reported, then the novelty of this paper may be judged to be low.
Response 1: It is definitely true. We incorrectly confused the terms "pilot data report" and "interim analysis report". Reference paper [7] reports planned interim analysis after enrollment of half planned subjects to verify the feasibility of the study.
Point 2: The authors need to describe the research method in more detail. The authors noted that there is a detailed protocol in reference (7), but only the randomization method is described in detail, and neither in the Clinical Trial gov registration nor in previous papers can be detailed about the protocol.
Response 2: Thank you for your suggestion. The manuscript has been supplemented mainly in the Method section. The randomization method, as well as patient enrollment, has been described in more detail.
Point 3: The allocation concealment needs to be described how it is performed, and the randomization method needs to be summarized in this paper. It also requires the details in what primary outcome is and what primary endpoint was. In the registration information for NCT03751566, the time frame for the primary outcome is six months after radiotherapy. However, the authors in this manuscript deals with radiotherapy information after three months and does not mention six months. It is necessary to reveal what the original primary endpoint was.
Response 3: Unfortunately, the registration information in the Clinical Trial gov contain mistake. The considered time frame for acute toxicity was three months in accordance with the manuscript methodology. The primary endpoint was pain level, with the primary outcome reducing acute toxicity in the acupuncture arm. The important points have been added to the manuscript.
Point 4: It is necessary to present the flowchart according to the CONSORT guidelines. There is no mention of drop out or withdrawl.
Response 4: We agree that the study flowchart is useful for readers and therefore has been added to the revised manuscript as Figure 1.
Point 5: Did all patients complete the treatment specified in protocol?
Any patients who have died or experienced SAE during the study period should be reported in flow chart.
Response 5: Yes, all patients complete the treatment according to protocol. Only two patients (one in each arm) dropped out during the follow-up after RT. No patients died or experienced SAE. This information has been added to the Methods section.
Point 6: According to that interim analysis, not only pain toxicity, but also skin toxicity and mucosal toxicity was presented. It is necessary to confirm whether the authors will report on this result in other studies.
Response 6: Thank you for your suggestion. We think that notes about other outcomes and other planned reports are useful for readers, and it has been added to the Introduction section.
Point 7: Based on the mean value of the average pain (it is also low in control group), some question is whether the pain in the RT of these patients is clinically significant. It is necessary to mention limitations on this.
Response 7: We fully understand your comment. The mean value may seem low and clinically nonsignificant. Nevertheless, the absolute value of the mean or median of the pain is affected by the analysis method. It is necessary to realize that the average pain level per time unit is shown in Figure 2B, i.e., it is affected by the length of the time window and by the fact that at the beginning of RT and later after the termination of RT patients often did not have pain. Moreover, an important factor in completing the treatment protocol is the strength of the acute toxicity of radiotherapy and their good symptomatic treatment. The goal of acupuncture is also to reduce acute toxicity, including pain, and increase the probability of completing treatment.
Reviewer 3 Report
Manuscript Review
JCM1120516
Although the findings are indeed very encouraging, future studies should absolutely include an active comparator arm, such as “sham acupuncture,” “healing touch (HT) or even just “time and attention.” It is a well-established fact that ANY additional care provided to patients tends to produce superior outcomes, so it is unclear how much of this effect is due specifically to “acupuncture” versus “additional care” to the patients. “Sham Acupuncture” would also be a fine active comparator arm.
The critique below may seem burdensome but is actually just some good housekeeping to make the paper stronger, and really may only require a long day of work (or two). This reviewer is not asking for any major revision, just some cleaning up and addition of various standard components to the paper
- The CONSORT format including the flow diagram should be followed. This would also help readers to understand in this article. Because it does not have a flow chart as Figure 1, I was not sure where the randomization occurred. Then the article states that the chemotherapy was not standardized to all participants. Some patients received chemotherapy as needed and some did not. How did this selective subject identification for chemotherapy affect the randomization? Is it really randomized equally and with equipoise? This is why this article needs a Flow Chart to understand exactly where randomization occurred, how patients were then stratified based on the type of RT (curative versus adjuvant), and then assignment to various chemotherapies for Control and Acupuncture arms (http://www.consort-statement.org/Media/Default/Downloads/CONSORT%202010%20Flow%20Diagram.pdf). It would also help to see visually and to clarify this confusing statement: Line 62/63: “The patients were randomized in ratio 1:1 into two arms – control and intervention (acupuncture) arm. Stratification was based on the type of RT (curative/adjuvant) and the application of concurrent chemotherapy.”
- Line 25, Line 29: In the Abstract, if 0.16 vs 1.36, etc. are representing Acupuncture versus the Control Group, respectively, then it should also report 0.8% versus 32.5%, 0% and 20.8%, for objective data, maintaining the same sequence rather than reversing to Control versus Acupuncture Group.
- Line 90: “Acupuncture was carried out by a certified physician (R.D.).” It would be of value to know where the certified physician was trained in acupuncture, how they were certified or licensed, what was the accrediting body, and how long the training period was.
- In Figure 1A and 1C, in the “Average Pain” for Acupuncture Group, it is interesting that the Average and Median are quite separated and would suggest that the data may not be normally distributed. Although this randomized open label trial is quite small, it is important to state or show whether the data were tested for a normal distribution, thus allowing the use of parametric versus non-parametric tests. This is also clearly seen in the Figure 1A Box and Whisker plots for the Acupuncture Group for “Average Pain” but not the “Worst Pain.”
- In Figures 1, 2 and 3 the authors use Box and Whiskers plots to show the IQR, but do not state the IQR values in the Results. Because of the small size of the study and obvious large variability, it is important for them to also include the IQR values in the results to give people a sense of the large variability in the data.
- Line 196-198: It is of note in Figure 4B that weak opiates were statistically significantly different between control and acupuncture groups. The authors should discuss this in the Discussion; why weak opiate use and not non-opiates and strong opiates—what is it about weak opiate use that acupuncture may be affecting—is it due to the type of pain they are experiencing? Worth commenting on.
- In the Discussion Section, limitations and potential bias is missing, including the fact that age and smoker distribution were significantly different between the experimental and control arms.
- Lines 229-234: The authors discuss “The other objective of this study was lowering the usage of analgesics,” however the authors provide no citations in this very important area of acupuncture research, treatment and outcomes. Please include a few high-level citations, such as the Cochrane Reviews that already exist examining the effectiveness of Acupuncture in the treatment of Cancer Pain, specifically looking at reduction of analgesic use. Perhaps these citations are already part of the systematic reviews on acupuncture reduction of self-reported cancer pain, but since this is highlighted as an important outcome measure, they should cite references that look at analgesic use specifically and the effects of acupuncture on analgesic use for cancer pain.
Author Response
Response to Reviewer 3 Comments
Although the findings are indeed very encouraging, future studies should absolutely include an active comparator arm, such as “sham acupuncture,” “healing touch (HT) or even just “time and attention.” It is a well-established fact that ANY additional care provided to patients tends to produce superior outcomes, so it is unclear how much of this effect is due specifically to “acupuncture” versus “additional care” to the patients. “Sham Acupuncture” would also be a fine active comparator arm.
The critique below may seem burdensome but is actually just some good housekeeping to make the paper stronger, and really may only require a long day of work (or two). This reviewer is not asking for any major revision, just some cleaning up and addition of various standard components to the paper
Point 1: The CONSORT format including the flow diagram should be followed. This would also help readers to understand in this article. Because it does not have a flow chart as Figure 1, I was not sure where the randomization occurred. Then the article states that the chemotherapy was not standardized to all participants. Some patients received chemotherapy as needed and some did not. How did this selective subject identification for chemotherapy affect the randomization? Is it really randomized equally and with equipoise? This is why this article needs a Flow Chart to understand exactly where randomization occurred, how patients were then stratified based on the type of RT (curative versus adjuvant), and then assignment to various chemotherapies for Control and Acupuncture arms (http://www.consort-statement.org/Media/Default/Downloads/CONSORT%202010%20Flow%20Diagram.pdf). It would also help to see visually and to clarify this confusing statement: Line 62/63: “The patients were randomized in ratio 1:1 into two arms – control and intervention (acupuncture) arm. Stratification was based on the type of RT (curative/adjuvant) and the application of concurrent chemotherapy.”
Response 1: We agree that the study flowchart is useful for readers and therefore has been added to the revised manuscript as Figure 1.
Point 2: Line 25, Line 29: In the Abstract, if 0.16 vs 1.36, etc. are representing Acupuncture versus the Control Group, respectively, then it should also report 0.8% versus 32.5%, 0% and 20.8%, for objective data, maintaining the same sequence rather than reversing to Control versus Acupuncture Group.
Response 2: Thank you for your suggestion. The abstract has been modified to be more clear.
Point 3: Line 90: “Acupuncture was carried out by a certified physician (R.D.).” It would be of value to know where the certified physician was trained in acupuncture, how they were certified or licensed, what was the accrediting body, and how long the training period was.
Response 3: The information about the training of the certified physician R.D. has been added to the methods section.
Point 4: In Figure 1A and 1C, in the “Average Pain” for Acupuncture Group, it is interesting that the Average and Median are quite separated and would suggest that the data may not be normally distributed. Although this randomized open label trial is quite small, it is important to state or show whether the data were tested for a normal distribution, thus allowing the use of parametric versus non-parametric tests. This is also clearly seen in the Figure 1A Box and Whisker plots for the Acupuncture Group for “Average Pain” but not the “Worst Pain.”
Response 4: You are absolutely right, the mean and median of average pain in Acupuncture arm are slightly different. The parametric test could be considered. However, with regard to the calculation of average pain level per time unit pain (i.e., AUC of the time course of pain for each pain), the 10-point rating scale of pain, and sample size, non-parametric methods were preferred. Using parametric methods (if the normality assumption would be met) would undoubtedly provide higher power of the tests, but the significance level of the differences between the arms would be similar. Altogether, we believe that non-parametric tests are the better option for this situation.
Point 5: In Figures 1, 2 and 3 the authors use Box and Whiskers plots to show the IQR, but do not state the IQR values in the Results. Because of the small size of the study and obvious large variability, it is important for them to also include the IQR values in the results to give people a sense of the large variability in the data.
Response 5: The IQRs were added into the manuscript where it was relevant to improve the informational value.
Point 6: Line 196-198: It is of note in Figure 4B that weak opiates were statistically significantly different between control and acupuncture groups. The authors should discuss this in the Discussion; why weak opiate use and not non-opiates and strong opiates—what is it about weak opiate use that acupuncture may be affecting—is it due to the type of pain they are experiencing? Worth commenting on.
Response 6: You are right; the observed statistically significant difference only for weak opioids is a slightly surprising result. We have been added some opinions about this to the Discussion section.
Point 7: In the Discussion Section, limitations and potential bias is missing, including the fact that age and smoker distribution were significantly different between the experimental and control arms.
Response 7: Thank you for your suggestion. The limitations of the study have been added to the discussion section.
Point 8: Lines 229-234: The authors discuss “The other objective of this study was lowering the usage of analgesics,” however the authors provide no citations in this very important area of acupuncture research, treatment and outcomes. Please include a few high-level citations, such as the Cochrane Reviews that already exist examining the effectiveness of Acupuncture in the treatment of Cancer Pain, specifically looking at reduction of analgesic use. Perhaps these citations are already part of the systematic reviews on acupuncture reduction of self-reported cancer pain, but since this is highlighted as an important outcome measure, they should cite references that look at analgesic use specifically and the effects of acupuncture on analgesic use for cancer pain.
Response 8: The reduction of analgesic use is certainly an important area of acupuncture research in cancer patients. Some published studies were added to the Discuss section. However, we were unable to find other references in English that would be close to our study design.
Round 2
Reviewer 2 Report
I believe that the manuscript has revised well by reflecting the comments.